# Adenosine and Adenosine Receptors: Advances in Atrial Fibrillation

**DOI:** 10.3390/biomedicines10112963

**Published:** 2022-11-17

**Authors:** Baptiste Maille, Nathalie Lalevée, Marion Marlinge, Juliette Vahdat, Giovanna Mottola, Clara Degioanni, Lucille De Maria, Victor Klein, Franck Thuny, Frédéric Franceschi, Jean-Claude Deharo, Régis Guieu, Julien Fromonot

**Affiliations:** 1Department of Cardiology, Timone University Hospital, 13005 Marseille, France; 2Centre for Nutrition and Cardiovascular Disease (C2VN), INSERM, INRAE, Aix Marseille University, 13005 Marseille, France; 3Laboratory of Biochemistry, Timone University Hospital, AP-HM, 13005 Marseille, France

**Keywords:** adenosine, adenosine receptors, atrial fibrillation, arrhythmia

## Abstract

Atrial fibrillation (AF) is the most common arrhythmia in the world. Because the key to developing innovative therapies that limit the onset and the progression of AF is to fully understand the underlying molecular mechanisms of AF, the aim of the present narrative review is to report the most recent advances in the potential role of the adenosinergic system in the pathophysiology of AF. After a comprehensive approach describing adenosinergic system signaling and the mechanisms of the initiation and maintenance of AF, we address the interactions of the adenosinergic system’s signaling with AF. Indeed, adenosine release can activate four G-coupled membrane receptors, named A_1_, A_2A_, A_2B_ and A_3_. Activation of the A_2A_ receptors can promote the occurrence of delayed depolarization, while activation of the A_1_ receptors can shorten the action potential’s duration and induce the resting membrane’s potential hyperpolarization, which promote pulmonary vein firing, stabilize the AF rotors and allow for functional reentry. Moreover, the A_2B_ receptors have been associated with atrial fibrosis homeostasis. Finally, the adenosinergic system can modulate the autonomous nervous system and is associated with AF risk factors. A question remains regarding adenosine release and the adenosine receptors’ activation and whether this would be a cause or consequence of AF.

## 1. Introduction

Atrial fibrillation (AF) is a multifactorial sustained cardiac arrhythmia, and it is now considered a real worldwide public health issue, affecting more than 50 million individuals [1,2]. Because this common arrhythmia, in clinical practice, is associated with major poor clinical outcomes, such as left ventricular dysfunction, heart failure or stroke, intensive research has been developed to understand the physiopathology of AF better. Despite the substantial progress that has been made in the detection and management of AF, the underlying molecular mechanisms associated with the onset of atrial fibrillation and its progression remain still unclear.

Among these molecular mechanisms, the implication of the adenosinergic system in AF has increased, since the accumulation of experimental data suggests that the increase in the adenosine blood level and the remodeling expression of the adenosine receptors might be part of the AF pathophysiology. Unfortunately, the adenosinergic system still has a Janus face in cardiac arrythmias, since adenosine can have both antiarrhythmic or proarrhythmic actions, along with adenosine receptors, which can lead to either profibrotic or antifibrotic effects [3,4]. Furthermore, whether adenosinergic system disturbances are the cause or the consequence of AF is not yet elucidated.

As the key to developing innovative therapies that limit the onset and progression of AF is to fully understand the underlying molecular mechanisms of AF, the aim of the present narrative review was to report the most recent advances on the potential role of the adenosinergic system in the pathophysiology of AF. After describing the adenosinergic system’s signaling and the pathophysiology of AF, this review focuses on the arrhythmogenic effects of adenosine and concludes with the association between the AF risk factors and adenosinergic system disturbance.

## 2. Adenosinergic System Signaling

### 2.1. Metabolism of Adenosine

Since Drury and Szent-Györgyi first observed the cardiac effect of adenine compounds in 1929 [5] and Burnstock proposed the concept of extracellular purinergic signaling in 1972 [6], adenosine has emerged as an important signaling molecule with pleiotropic actions, including effects on the cardiovascular system [3,4,7]. This ubiquitous purine nucleoside comes mainly from ATP dephosphorylation when tissue energy requirements increase, such as during hypoxia, ischemia or inflammation [7]. The methionine cycle can contribute to intracellular adenosine formation in the heart by the hydrolysis of S-adenosylhomocysteine [8]. However, ATP hydrolysis is considered the main source of adenosine secondary due to the extracellular dephosphorylation cascade of ATP by the successive action of a membrane-anchored ectonucleoside triphosphate diphosphohydrolase-1 (CD39) and the ecto-5’-nucleotidase (CD73) [9,10,11] (Figure 1). Consequently, all situations associated with a massive release of adenine nucleotides (e.g., cell damage, platelet aggregation, and vascular shear stress) can rapidly increase the level of adenosine from 0.4–0.8 µmol/L under resting physiological conditions to more than 1.0 µmol/L in pathologic conditions, such as neurally mediated syncope or cardiogenic shock [12,13].

Blood concentrations of adenosine are limited in time and space by its catabolism and reabsorption by balancing nucleoside transporters (ENTs), particularly in red blood cells [14,15,16]. Adenosine deaminase (ADA), which is present in most cells and on the surface of mononuclear cells (MC-ADA), converts adenosine to inosine, which is metabolized to uric acid, its final catabolite. Moreover, adenosine kinase can phosphorylate intracellular adenosine and convert it into adenosine monophosphate (AMP) [17]. Consequently, the half-life of adenosine is short (10–30 s) due, notably, to the fact of its rapid uptake by erythrocytes and its deamination by adenosine deaminase [18,19], and its rate of turnover rate is 1.5 nmol/mL/min [17].

During hypoxia, ischemia or inflammation, ATP metabolism is modified leading to increased production of adenosine, which is associated with a decrease in its rephosphorylation. Inflammation or hypoxia can modulate the adenosinergic system thanks to the transcription factor NF-κB, which increases the transcription of adenosine receptors and hypoxia-inducible factor-1α (HIF-1α) genes [20,21,22,23]. In hypoxic conditions, the proteasomal degradation of prolyl-hydroxylase allows for the nuclear translocation of the transcription factor HIF-1α to induce the transcription of numerous genes, including those for adenosine receptors and CD73 and NF-κB, and to repress the transcription of ENT and adenosine kinase genes [24,25]. Thus, increased adenosine plasma concentrations have been well reported in ischemic or inflammatory diseases, such as coronary artery or ischemic heart diseases [26,27] and pulmonary inflammation [28,29].

### 2.2. Adenosine Receptors and Their Cardiac Effects

Adenosine receptors are subdivided in four G protein-coupled receptors, named A_1_ (A_1_R), A_2A_ (A_2A_R), A_2B_ (A_2B_R) and A_3_ (A_3_R) receptors, which are classified according to their primary sequence, their associated G-protein and their pharmacological profile (binding affinity of agonists and antagonists) [30,31]. A_1_R and A_3_R are coupled to the Gi/o proteins leading to the inhibition of adenylyl cyclase (AC), whereas A_2A_R and A_2B_R are associated to Gs proteins, which stimulate the adenylyl cyclase and increase the intracellular production of cyclic-AMP (cAMP). A_1_R and A_2A_R have a high affinity for adenosine with a higher binding affinity of A_1_R. A_2B_R and A_3_R have a lower affinity for adenosine [31,32].

Adenosine receptors are widely distributed in various cells and tissues, including the cardiovascular system [33,34]. All four adenosine receptors have been described in the heart, with distributions varying regionally. Their activations by adenosine have major effects on cardiac function by modulating the sympathetic tone and the conductance of potassium and the calcium current [34,35]. Cardiac A_1_R is highly expressed in the atrial myocardium, the sinoatrial node (SAN), the atrioventricular node (AVN) and the His–Purkinje system and presynaptically on adrenergic nerve varicosities but have a lower expression in ventricular myocytes [36]. The activation of A_1_R by adenosine induces direct negative chronotropic and dromotropic effects [37] and an indirect anti-β-adrenergic action which can antagonize the positive inotropic effect of catecholamines [38,39,40] (Figure 2). This is especially useful as a cardioprotection in ischemia-reperfusion injury [41,42]. A_2A_Rs are particularly expressed in vessels and coronary arteries with a vasodilator effect [43,44], whereas their density is moderate in ventricular and atrial tissues, leading to the contraction of cardiomyocytes [45,46,47]. Opposite to A_1_R, A_2A_R directly enhanced the inotropic effect of cardiomyocyte by increasing the cytosolic Ca^2+^ level and the Ca^2+^ sensitivity of the myofilaments [48]. In the same manner, neither deletion of the A_2A_ or A_2B_ receptors nor the brief activation of these receptors impacts ischemia-reperfusion injury [49]. However, in A_2A_R or A_2B_R knockout mice, the beneficial effects of A_1_R stimulation on post-ischemic function and infarct size were reduced [49]. A_2B_Rs are expressed in coronary arteries and mediates coronary vasodilation, with a minor and supportive role compared to A_2A_R [50]. The expression of A_2B_Rs has also been reported in atrial and ventricular cardiomyocytes [35,47]. A_2B_Rs’ activation on cardiac fibroblasts reduces collagen and total protein synthesis suggesting that adenosine may protect against cardiac fibrosis [51]. Expressed in the coronary arteries, A_3_Rs’ activation mediated coronary vasodilation [52] and can also induce cardioprotection, since their activation by a high-selective agonist, IB-MECA, reduced myocardial infarct size [53].

Previous studies have shown that A_1_R and A_2A_R were involved in physiological heart rate regulation at a 10^−5^ to 10^−3^ M concentration [54]. A_1_R induced bradycardia by depressing the sinoatrial node and slowing the atrioventricular conduction [55,56]. Since mice KO for A_2A_R exhibit an increase in heart frequency, this suggests that A_2A_R activation also leads to bradycardia [57]. This regulation of the heart rate appears to be partly due to the modulation of the sympathetic tone [58] and suggests a collaborative effect between the different receptor subtypes for cardioprotection [42,59].

### 2.3. Molecular and Ionic Bases of Adenosine Effects in Cardiac Electrophysiology

The electrophysiological action of adenosine in the heart is regionally variable and dependent on the underlying ionic current population, which differ between species [41]. Irrespective of the species, adenosine exerts its cardiac electrophysiological actions mainly through the modulation of potassium, sodium and calcium currents by cardiac A_1_ and A_2A_ receptors’ activation [60,61].

#### 2.3.1. Effects of Adenosine on Ionic Currents

Adenosine is believed to induce a cAMP-dependent (indirect effects) anti-β-adrenergic action by a dual mechanism: the activation of the presynaptic A_1_R limits the release of norepinephrine [62] and the activation of postsynaptic A_1_R blocks the effects of catecholamines by inhibiting adenylyl cyclase activity and then reducing intracellular cAMP levels. This indirect effect subsequently decreased the catecholamine-induced calcium inward current through L-type calcium channels (ICa,L) [55] and the sodium inward current (“funny” current (If) or pacemaker current) through hyperpolarization-activated cyclic nucleotide-gated (HCN) channels [63]. Nevertheless, it is still considered that the main cardiac effect of A_1_R is direct and cAMP-independent by modulating an inwardly rectifier potassium current, which is activated by both adenosine and acetylcholine (IKAch,Ado), herein further referred to as IK,Ado [55,64].

The activation of A_1_Rs enhances the IK,Ado thanks to G-protein-coupled, inwardly rectifying K+ (GIRK) channels [65]. After the Gi-protein activation, the G_i_α subunits enhance the gating of the GIRK channels, in the same manner as acetylcholine on the muscarinic-2 receptor, which causes a vagally mediated negative chronotropy upon on atrial pacemaker activity [65,66]. The GIRK channels aim to maintain the potassium equilibrium potential (EK+ = −90 mV) and modulate its conductance according to the membrane potential. As other inward rectifiers, they conduct larger inward currents when the membrane potential is negative to the EK+ than outward currents when the membrane potential is positive to EK+ [67,68,69]. The GIRK subunits (GIRK1–4 or Kir 3.1–3.4) are associated with A_1_R in the heart and, more particularly, in the SAN and atria [70,71,72]. Thus, adenosine causes a GIRK-induced potassium outward current leading to resting membrane potential (RMP) stabilization or hyperpolarization, a reduced rate of diastolic depolarization and a shortened action potential duration (APD) and an effective refractory period (ERP) [55]. Complementary to the GIRK-induced IK,Ado, A_1_Rs regulated the cAMP level by the inhibition of the adenylate cyclase activity through the G_i_α-subunit. Therefore, the cAMP-dependent protein kinase A (PKA) activity and the PKA-dependent phosphorylation of Ca^2+^-handling proteins are reduced, which modulates Ca^2+^ homeostasis and reduces isoproterenol-stimulated ICa,L [55].

On the contrary, like other G_S_-protein coupled receptors, A_2A_R stimulation can activate a cAMP-dependent pathway modulating Ca^2+^ handling. However, the precise mechanism of the A_2A_R-induced transduction signal remains controversial. Indeed, some authors report that the stimulation of A_2A_Rs by a specific agonist (i.e., CGS 21680) did not alter the cAMP level in rat ventricular cardiomyocytes [73], whereas, more recently, others showed that CGS 21680 increased the cAMP content in ventricular cardiomyocytes of transgenic mice overexpressing human A_2A_R [74].

Moreover, in rat ventricular cardiomyocytes, the A_2A_R-induced inotropic effect was associated with a Ca^2+^-dependent pathway, but it was also considered as predominantly mediated by a Ca^2+^-independent inotropic pathway, which may rely on an increased myofibrillar Ca^2+^ sensitivity [75]. In human right atrial preparations, A_2A_Rs’ stimulation by CGS 21680 induces a PKA-mediated increase in the frequency of spontaneous calcium release from the sarcoplasmic reticulum (SR) without a significant increase in the ICa,L amplitude [46].

In the canonical Ca^2+^-dependent pathway, the PKA-dependent phosphorylation might increase the activation of L-type Ca^2+^ channels (LTCC) initiated by membrane depolarization. Following LTCC opening, the inward Ca^2+^ current (ICa,L), in turn, triggers a Ca^2+^-induced Ca^2+^ release (CIRC) from the sarcoplasmic reticulum via the phosphorylated cardiac ryanodine receptors (RyR2) and which is responsible for numerous spontaneous Ca^2+^ release events (Ca^2+^ spark and Ca^2+^ waves) during systole [46,76]. This can lead to the positive inotropic effect of A_2A_R stimulation in atria and ventricles through the Ca^2+^-induced myofibrilla contraction responsible for the cardiac electromechanical coupling [48]. The structural organization of cardiomyocytes into specific microdomains (i.e., cardiac dyads) favors this cardiac excitation–contraction coupling [77,78]. This is supported by the regional variation of the phosphorylation state of the LTCC [79] and the expression and the activity of the Ca^2+^ handling proteins [78,79,80].

Because an appropriate cardiac function requires a tight regulation of the Ca^2+^ content at the end of the systole to hasten the relaxation (positive lusitropic effect), it is interesting to note that in mice overexpressing human A_2A_R, the stimulation of A_2A_R increased the phosphorylation states of phospholamban (PLN) and the inhibitory subunit of troponin (cTnI) [74,81]. The PKA-dependent phosphorylation of cTnI hastens the dissociation of Ca^2+^ from troponin C, allowing for a faster relaxation rate [82] The phosphorylation of PLN de-inhibits the activity of the sarcoplasmic reticulum Ca^2+^ ATPase (SERCA) pump which, in turn, reuptakes cytosolic Ca^2+^ and leads to a lower Ca^2+^ content during the diastolic phase [83]. During diastole, the Ca^2+^ extrusion is also mediated by the Na^+^-Ca^2+^ exchanger (NCX: stoichiometry of 1Ca^2+^:3Na^+^) allowing for cell relaxation. Indeed, the beat-to-beat electromechanical coupling stability is determined by the ability of the myocyte to maintain equilibrium between the SR calcium’s uptake and release [84].

Therefore, the effects of adenosine in the nodal cells (i.e., SAN and AVN), the conductive tissues and in the working myocardium are largely based on the modulation of adenylate cyclase activity, which induces cAMP-dependent PKA signaling, leading to Ca^2+^ handling protein phosphorylation, and to a GIRK-induced potassium outward current (IK,Ado) (Figure 3).

#### 2.3.2. Adenosine Effects on the Action Potential in Nodal Cells and Working Cardiomyocytes

The action potential of the sinoatrial node cells is characterized by a more depolarized (−60 mV) and more labile resting membrane potential than contractile cardiomyocytes due to the almost lack of Kir2-encoded inwardly rectifier potassium currents (IK1) [85,86]. After a previous action potential, when the membrane potential reaches the maximum diastolic membrane potential, the HCN channels’ opening allows for the inward Na+ “funny” current (If), which contributes to the automaticity. This spontaneous diastolic HCN-mediated depolarization drives the membrane potential to the action potential threshold (~−40 mV) [87]. This threshold triggers the opening of voltage-dependent L-type Ca^2+^ channels, which induce a low slope of depolarization (phase 0). Lastly, while the Ca^2+^ channels close, the rapid and slow delayed rectifier current (IKr, IKs) induces outward K^+^ currents, which are responsible for the repolarization (Figure 4A). Even if the IK1 current is very weak in sinoatrial cells, others inward rectifiers, including the acetylcholine/adenosine-sensitive and ATP-sensitive inward rectifiers, may also modulate cardiac excitability. The stimulation of A_1_R by adenosine on a sinoatrial node induced a time- and voltage-dependent negative chronotropy under basal (activation of IK,Ado) and β-adrenergic stimulated conditions (inhibition of If and ICa,L) [55]. The direct activation of IK,Ado induces a hyperpolarization of sinoatrial nodal cells and decreases the slope of the diastolic depolarization and the repolarization duration. Because the sinoatrial node is under the influence of the sympathetic nervous system, the anti-β-adrenergic effect secondary to A_1_R activation limits the cAMP-dependent currents (If and ICa,L), which further decreases the slope of the diastolic depolarization [41,60,63] (Figure 4A).

In an atrioventricular node, adenosine induces mainly a direct negative dromotropic effect. Whereas the nodal-his bundle cells are insensitive to adenosine, adenosine decreases the action potential duration in atrio-nodal cells. The most important action occurs on nodal cells, since the concentration-dependent effect of adenosine can decrease the amplitude and duration of the action potential and abolish the action potential at high concentrations of adenosine [41,60,68].

Conversely to nodal cells, atrial and ventricular myocardium are fast response cardiac tissues with Na^+^-dependent depolarization (INa) through the voltage-gated sodium channels Na_v_1.5 [88]. Because of the smaller density of the inward rectifier K^+^ current (IK1) in atrial cells, the human atrial action potential typically demonstrates a triangular morphology and a more depolarized resting membrane potential (~−70 mV) compared to the spike-and-plateau shape of ventricular cardiomyocytes (−90 mV) [89]. The initiation of the atrial action potential is permitted by a rapid inward Na^+^ current (INa), followed by an inward Ca^2+^ current due to the opening of L-type Ca^2+^ channels (ICa,L), which finally close during repolarization, while inward rectifier K+ channels open (Figure 4B). Even if a specific A_2A_R stimulation is associated with a positive inotropy in atria, the main physiological effect of adenosine is mediated through A_1_R due to the atrial distribution of the adenosine receptors, their affinity for adenosine and the sympathetic regulation of atrial myocardium. Contrary to sinoatrial cells, the direct activation of IK,Ado slightly hyperpolarized the resting membrane potential of atrial cells, since this resting voltage is near the potassium equilibrium potential and the GIRK-induced K+ outward current is reduced [90]. However, adenosine shortens the action potential duration and decreases the refractory period thanks to the IK,Ado outward current, which increases the potassium repolarization flux (Figure 4B). It is noteworthy that under adrenergic stimulation adenosine can also reduce the cAMP-dependent effect on ICa,L in atrial cells [55,60,91] Moreover, this shortening of the atrial action potential by the direct and fast onset of IK,Ado may reduce the time needed for the slowed onset of cAMP-dependent pathways, leading to cardiac electromechanical coupling and thereby depressing the twitch amplitude and decreasing the atrial contractile force [55].

On ventricular cardiomyocytes, adenosine action does not cause any significant effect on the resting membrane potential, action potential duration or amplitude [55].

Using these characteristics, the major diagnosis interest and clinical use of adenosine is due to the fact of its action on an atrioventricular node. It confers to adenosine its diagnosis properties used in supra ventricular tachycardia [60,61]. While shortening the action potential duration, adenosine can also elicit a post-repolarization refractoriness prolongation [92]. Thus, a high exogenous dose of adenosine can lead to an atrioventricular block.

## 3. Pathophysiology of Atrial Fibrillation

Atrial fibrillation (AF) is the most common pathologic tachyarrhythmia, defined by the absence of distinct repeating P-waves, irregular atrial activations on surface ECGs and irregular R-R intervals. At the beginning, episodes of sinus rhythm are punctuated by periods of arrhythmia (paroxysmal AF). AF progression leads to more frequent and longer episodes until they last more than 7 days (persistent AF). After one year, in the absence of spontaneous or medical cardioversion, which allow a return to sinus rhythm, AF is then considered as a “long-standing persistent” AF. Finally, AF can be defined as a “permanent” AF when the therapeutic attitude is to consider that no further attempts to control sinus rhythm will be undertaken [1]. AF can lead to a large number of symptoms, including palpitation, breathlessness, heart failure, dizziness or syncope, asthenia or chest pain. In addition to symptomatic forms, asymptomatic AF is called subclinical AF [1]. The AF consequences are multiple. The increasing risk of thromboembolic events is the best known [93]. However, more evidence has also been published concerning the increasing risk of heart failure [94], dementia [95], depression [96] and impaired quality of life [97].

The pathophysiology of AF is supported by anatomical and electrophysiological changes in atria for which all of the exact molecular mechanisms are still unknown. Understanding of this complex AF pathophysiology is essential to identify the underlying molecular mechanisms. According to the Coumel triangle, AF pathophysiology first depends on trigger factors, which are rapid focus firing activity, initiating the arrhythmia. Then, the arrhythmogenic substrate are electrophysiological, mechanical and anatomical characteristics of the atria that sustain AF. Further, multiple neuro-humoral dynamic modulators also interfere with the initiation, progression and termination of AF. Therapeutic targets for a rhythm control strategy should consider all aspects of this classification [98]. Unfortunately, the patterns of AF are only partially correlated with the progression of the underlying mechanism of AF [99]. It is thought that the first episodes of paroxysmal AF are usually based on triggering activity. Then, according to the underlying fibrosis progression, functional drivers of AF begin to anchor on the structural remodeling area, and AF tends to become persistent [99].

### 3.1. Atrial Fibrillation Triggers

Repeated atrial ectopias are known to initiate AF and are considered as triggers of AF [100]. They can be localized within the pulmonary veins (PVs) and, less frequently, in others atrial areas (non-PV triggers). Among patients affected by AF, 20% of them exhibit non-PV foci [101], which can be predicted by female gender (odds ratio: 2.00), left atrial enlargement (odds ratio: 2.34) and AF episode prolongation [102,103]. Mapping studies have identified and localized a discrete clustered anatomical area corresponding to non-PV foci in the inferior mitral annulus, the posterior left atrium, the interatrial septum particularly at the fossa ovalis/limbus region, the crista terminalis and Eustachian ridge, the coronary sinus, and the superior vena cava [103].

PV ectopias originate all along the myocardium sleeve of PV and present unpredictable firing with various, intermittent and delayed conduction to the left atrium. They are defined as focal discrete sites of early and centrifugal activation [104]. Compared to the atrial cells, the PV cardiomyocytes have specific action potential properties that predispose to arrhythmogenesis [105]. Indeed, PV cells have a higher resting membrane potential, a lower amplitude of the action potential, a smaller maximum phase 0 upstroke velocity and a shorter action potential duration (Figure 5).

PV ectopia beats are mainly related to Ca^2+^ handling abnormalities leading to cytosolic Ca^2+^ overload. Indeed, the Ca^2+^ overload leads to the diastolic leak of Ca^2+^ from sarcoplasmic reticulum (SR), called Ca^2+^ sparks and Ca^2+^ waves. The SR Ca^2+^ released is supported by the ryanodine receptors type-2 (RyR2), whereas Ca²^+^ storage is endorsed by the sarco-endoplasmic reticulum Ca²^+^ ATPase (SERCA2). Dysfunction of one or the other can induce diastolic Ca²^+^ leaks [106,107]. Consequently, the cytosolic Ca^2+^ overload activates the Na^+^-Ca^2+^ antiporter (NCX) to remove Ca^2+^ from cells with a stoichiometry of 3Na^+^:1Ca^2+^ [108]. The associated inward NCX current (I_NCX_) can induce delayed afterdepolarizations (DADs) [109], which can be sufficient to meet the threshold cell membrane voltage and to trigger an action potential [110].

Secondary to PV ectopy, firing can be related to DAD due to the fact of abnormal diastolic spontaneous calcium release [111]. Induced electromechanical coupling beat-to-beat alternans may especially precede and induce AF by an increasing dispersion of refractoriness [112,113]. Furthermore, atrial myocytes at the entrance of the PV have abrupt changes in their fiber orientation, which locally reduce the conduction velocity. This anisotropy leads to further increased dispersion of refractoriness and favor reentry [114,115]. Such abnormal depolarization and subsequent firing activity at the junction between the PVs and left atrium are responsible for the occurrence of arrhythmias [116] and promote reinduction of AF [117].

### 3.2. Mechanisms of AF Perpetuation

Two major mechanisms of perpetuation supported by anatomical and/or electrophysiological changes are described by the multiple wavelets hypothesis [98,118] and the localized (focal or reentrant) AF drivers.

The multiple-wavelet phenomenon hypothesizes that multiple random wavefronts propagate through the atria until depolarizable tissue is available. Secondary to AF, atrial enlargement increases the critical atrial mass that is required to self-perpetuate this mechanism [119]. Within a same critical atrial mass, a short refractory period and delayed conduction velocities increase the theoretical number of wavelets. Furthermore, as the number of wavelets increases, the chance of spontaneous AF termination is lower [120,121]. This phenomenon is supported by the fact that larger mammals are more likely to exhibit AF [122]. Furthermore, an effective left atrium conduction size enlargement has recently been associated with AF vulnerability in persistent AF [123].

AF drivers are defined as a localized source of fast and repetitive activity during AF episodes from which activation propagates and breaks down into fibrillation of the rest of the atria [124]. The drivers are considered to be focal when the wavefront activation originates from a focal site with centrifugal activation and as reentrant when the waves fully and continuously rotate around an anatomical or functional pivot point [125]. As ectopias, because of the electrophysiological characteristics of PV cells and the brutal change in PV orientation fiber, the drivers are mostly located in the PV antral and adjacent regions [114,125]. With a longer AF duration, the complexity of the AF drivers increases, and they are located at extra-PV sites [125].

Anatomical reentries are defined by the presence of an unidirectional slow conduction area or block resulting in a fixed cycle length and localization circuit. Atrial fibrosis favors these slow conduction areas [126]. The nonuniform anisotropic conduction within a fibrosis area is an important substrate for reentrant tachycardia [118]. Functional reentries are defined by an absence of an underlying substrate and/or anatomical obstacle. Figure 6 illustrates the formation of a functional reentrant circuit due to the heterogeneity of the atrial potential duration at the PV junction. Then, the functional reentry can be schematized as a central refractory area maintained by centripetal waves moving around [127] or a propagation of spiral wave reentry or “rotor” around a core area at which the depolarization and the repolarization curves join each other [128]. In these models, the size of the reentrant circuit is dependent on the conduction velocity and refractory period (minimal time allowing for re-excitation). As the conduction velocity increases and the refractory period decreases, the circuit size decreases and the potential number or circuit within the same atria increases [128]. A recent computational modeling study supported an increased reentry inducibility in the case of shortening the PV action potential duration compared to the left atrium action potential duration and slowing the conduction velocity within the PV toward the PV/ left atrium junction [114]. Other authors have also confirmed that a shortened action potential duration and effective refractory period associated with a hyperpolarization increase the rotors’ stability [129,130]. The underlying molecular mechanisms in paroxysmal AF patients are suspected to be related with the heterogeneity of the potassium inward rectifier channels’ (IK1 and IKACh) expression and conductance within the left atria [131]. This heterogeneity could promote a reentry mechanism by the heterogeneous shortening of the action potential durations. The induced hyperpolarization of atrial cardiomyocytes also removes the voltage-dependent sodium current inactivation and increases the sodium channel availability [132]. This leads to the acceleration and stabilization of the rotors.

Both mechanisms of reentry can be present within the same atrium. There is evidence of intermittent and spatially unstable drivers anchored to the structural heterogeneities of the atrial fibrosis area, which is considered to be the major mechanism of human AF perpetuation [118].

### 3.3. Substrate of Atrial Fibrillation and Atrial Cardiomyopathy

The AF perpetuation is caught in a vicious circle. The triggers and perpetuation mechanisms are dependent on multiple underlying electrical remodeling and atrial fibrosis which are also induced by AF. This vicious circle is supported by the fact that a complete electrical reverse remodeling occurred within weeks after medical or electrical cardioversion [121,133]. Then, the structural alterations can also partially or completely reverse within months after cardioversion [121,133]. Consistently, in 2020, the guidelines of AF management considered that all cardiovascular risk and comorbidities contributing to the development of an abnormal atrial substrate should be strictly controlled in order to limit AF progression [1].

Atrial cardiomyopathy has been defined as “any complex of structural, architectural, contractile or electrophysiological changes affecting the atria with the potential to produce clinically relevant manifestations” [134]. Thus, any initial electrostructural changes secondary to AF should be considered as an atrial cardiomyopathy. As a consequence, AF should be considered as a risk factor of atrial cardiomyopathy. All other atrial aggressions (i.e., diabetes, hypertension, aging, heart failure, valvular diseases, amyloidosis, granulomatosis and inflammatory infiltrates) primarily responsible for atrial changes, such as fibroblast or noncollagen deposits, are risk factors of AF. In this context, AF is considered as a marker of underlying atrial cardiomyopathy.

### 3.4. Pejorative Modulators of Atrial Fibrillation

The onset and the termination of AF episodes are likely related to dynamic modulators, which interact with both the trigger and substrate of AF. They exhibit multiple timescales and sites of action, including immediate and long-term electrophysiological atrial changes, ion channel conductance and homeostasis, and local and general inflammation status as well as fibrosis promotion [109]. Nonexhaustive modulators of AF are the autonomic nervous system, obstructive sleep apnea, electrolytes and plasma glucose levels, gastroesophageal reflux disease, ischemia, inflammation, thyroid diseases and advancing age [109].

## 4. Arrhythmogenic Effects of the Adenosinergic System

### 4.1. Adenosine Level and Expression of Adenosine Receptors in AF Patients

High adenosine plasma levels have been found in the left atria of patients during episodes of paroxysmal AF and in persistent AF [135]. The adenosine plasma concentrations then normalized after spontaneous or electrical cardioversion in sinus rhythm [135]. Moreover, the adenosine plasma concentrations in peripheric blood circulation were also higher in permanent AF compared to paroxysmal AF and controls [135]. The high adenosine plasma concentrations could be attributed to peripheral hypoxemia caused by the decrease in the left ventricular output in AF [135,136]. Interestingly, AF prolongation episodes were inversely correlated with the decrease in the atrial bioenergetic level [136]. Moreover, adenosine deaminase (ADA) activity is decreased in patients with paroxysmal AF contrary to patients with isolated dilated left atrium without AF [135,137]. Therefore, it was hypothesized that decreasing the ADA activity could be a further step before AF occurrence [137]. Nevertheless, further studies are still needed to understand whether a reduction of the ADA activity is the cause of the adenosine plasma level increase.

A high adenosine plasma concentration could also be a consequence of energy use in specific underlying cardiovascular conditions, including hypertension [138,139], chronic heart failure [13,140] or vagal syncope [141,142]. These are especially known to be AF risk factors. Interestingly, AF initiation has been described during the strong release of adenosine or the use of extrinsic adenosine injection [143,144]. Thus, AF would be a consequence of adenosine release.

The expression of adenosine receptors is modified in AF. Our team recently described the upregulation of A_2A_R and A_2B_R expressions in the left and right atrium in postoperative AF [47]. This overexpression was significantly higher in the left atrium than in the right atrium [47]. Li et al. also showed the upregulation of A_1_R in AF, especially in the right atrium [72]. In patients affected by AF, the overexpression of A_2A_ and A_2B_ receptors, as was also described in peripheral blood mononuclear cells (PBMCs), was correlated with the atrial expression of adenosine receptors, suggesting that the peripheric status of the adenosinergic system could be a mirror of the underling atrium disease [135,137].

### 4.2. Implication of A_1_ Receptors in AF

In Langendorff preparation of rat hearts, the injection of a specific A_1_R agonist (CCPA) produced a profound negative chronotropic effect, whereas the selective A_1_R antagonist (PSB36) produced a nonsignificant positive chronotropic effect [145]. The increased concentration of both agonist and antagonist produced runs of repetitive atrial ectopy. This clinical effects appears to be mainly driven by a shortening of the action potential and effective refractory period durations, resulting in an increasing AF susceptibility due to the A_1_R activation [145]. Indeed, the activation of A_1_R through IK,Ado modulation induces a resting membrane potential hyperpolarization, a reduction of the action potential duration and a shortened effective refractory period (ERP) on atrial cardiomyocytes [65] (Figure 7).

In a canine model, Datino et al. showed that adenosine enhanced a larger IK,Ado current in PV cardiomyocytes compared to left atrium cells, which resulted in the heterogeneity of the hyperpolarization. Conversely, the ERP of PV cardiomyocytes and left atrial cells both similarly decreased [146]. Hyperpolarization activates hyperpolarized-activated cyclic nucleotide-gated (HCN) channels [147,148]. The activation of HCN results in a Na^+^ inward current (If or pacemaker current), which is usually specifically located within the sinoatrial node cells. However, their presence has been demonstrated within PV cardiomyocytes [149].

Following PV ectopic activity, a shortened effective refractory period allows for PV firing. Furthermore, a similar decrease in the ERP between PV and LA cells does not modify the underlying heterogeneity of the ERP between both structures [105]. Hyperpolarization also increases sodium channel availability [132], which increases the velocity of the phase 0 depolarization within PV cardiomyocytes [146] and enhances the local rotors’ initiation and prolongation [128].

Observations by Li et al. support this concept. They showed that adenosine-induced APD shortening was more significant in the right atria than in the left atria. They also showed that adenosine-induced atrial fibrillation was mainly associated with reentrant drivers localized in the lateral right atria, which is associated with a higher expression of A1R and GIRK4 [72]. In this study, the use of ex vivo atrial preparation from a transplanted failing heart avoided the analysis of the compounding influence of the autonomic nervous system. However, for its ability to stabilize the AF rotors, adenosine was proposed by the authors as a tool to detect the location of possible AF foci during ablation procedures [72]. Recently, Pope et al. confirmed that adenosine injection in persistent AF shortened the mean AF cycle length and increased the density of the rotating wavefronts throughout the left atrium [130]. Unfortunately, the utility of these rotor ablations in AF termination is uncertain, because the increased density of the rotors may just reflect a more excitable state of the left atrium [130].

Interestingly, in AF, the reduced supply of oxygen in localized atrial regions of a canine model showed a profound conduction slowing within the ischemic zone, which promoted reentry mechanisms [150]. Furthermore, during hypoxia, HIF-1α activation increases adenosine availability and slows the adenosine metabolism rate [24]. This can support the hypothesis that the expression and signal transduction of adenosine receptors could differ between cardiomyocytes in several regions and participate in electrophysiological atrial heterogeneity.

The brady-tachycardia syndrome associates a sinus bradyarrhythmia with tachyarrhythmia, such as AF, which is a hallmark of heart failure [151]. Chronic heart failure is associated with A_1_R overexpression [140]. In a canine model, the increased A_1_R expression in a sinoatrial node and atrial cardiomyocytes was associated with adenosine-induced sino-atrial node conduction abnormalities and shorter atrial repolarization, which increase the risk of AF [152]. These effects are abolished by a selective GIRK channel blocker [71]. In clinical practice, these effects support the clinical use of theophylline, a nonselective adenosine antagonist, to prevent the occurrence of congestive heart failure in patients with brady-tachycardia syndrome. However, the high rate of side effects associated with global A_1_R inhibition limits its practical use [153].

### 4.3. Implication of AR in the Remodeling of Calcium Handling

In atrial cardiomyocytes, a reduced L-type Ca2^+^ current (ICa,L) density [154] and increased spontaneous Ca^2+^ release from the sarcoplasmic reticulum through the ryanodine receptor [110] are implicated in the cytosolic Ca^2+^ overload responsible for delayed afterdepolarization (DAD), which promotes AF [116,117]. These effects on calcium remodeling are consistent with adenosine receptor signaling. Despite A_1_R activation inducing a c-AMP-dependent decrease in ICa,L, it has limited or no effect on the action potential duration [155]. Indeed, most studies report the major role of A_2A_R in Ca^2+^ handling remodeling in AF.

Interestingly, the evidence for proarrhythmic effects of A_2A_R was reported in transgenic mice with cardiac overexpression of human A_2A_R [81]. In human right atrial cardiomyocytes, the selective activation of atrial A_2A_R can increase the frequency of a PKA-dependent spontaneous Ca^2+^ release from the sarcoplasmic reticulum without modulating L-type Ca^2+^ current [46]. This was confirmed in right atrial samples of patients with AF, where A_2A_R overexpression is associated with an increased phosphorylation state of the ryanodine receptor, an increase in Ca^2+^ spark and wave frequency and a subsequent increase in the spontaneous I_NCX_ frequency [76]. However, it is noteworthy that A_2A_R activation in these right atrial samples did not modulate the Ca^2+^ content of sarcoplasmic reticulum, the amplitude of ICa,L and the I_NCX_. However, the remodeling of Ca^2+^ handling associated with A_2A_R activation by adenosine has been reported to alter the ability of human atrial myocytes to maintain a uniform beat-to-beat response [156]. In parallel to the beat-to-beat alteration, Molina et al. confirmed that A_2A_R activation was linked to an increased spontaneous Ca^2+^ release, which can elicited DAD by the arrhythmogenic transient inward current (Iti) (Figure 8).

### 4.4. Modulation of Atrial Fibrosis by A_2B_ Receptors

The activation of A_2B_R has an important role in cardiac fibroblast homeostasis, but the role of A_2B_ in fibrosis is controversial, since both profibrotic [157,158,159,160] and antifibrotic effects are reported [161,162,163,164,165]. In cardiac fibroblasts, the increase in cAMP production following A_2B_R activation plays an essential role in the inhibition of angiotensin-II-induced collagen production [162]. However, activation of the A_2B_R increases the production of collagen and increases the release of IL-6 in human cardiac fibroblasts, resulting in a profibrosis state [160,166]. These opposite results may be the consequence of an activation of the A_2B_R that is time dependent on the A_2B_R, with early protective effects in acute post-myocardial ischemia and a later inappropriate healing runaway [167].

Until there, scarce data are published on implication of A_2B_R in atrial fibrillation and fibrosis proliferation. Recently, our group demonstrated an overexpression of A_2B_R in AF, especially within the left atrium [47]. In the same manner as A_2B_R antagonist reduces ventricular remodeling and tachycardia vulnerability after myocardial infarction [167], A_2B_R may be implicated in fibrosis modulation in atrial fibrillation.

Other ARs have also been implicated in profibrotic action or reduction of myocardial remodeling. Similarly, activation of A_2A_R could also reduce the angiotensin-II-induced collagen production, especially in the case of hypertension [168]. Nevertheless, it has also been associated with profibrotic effects in other organs, such as the liver and skin [169]. However, other AR expressions in cardiac fibroblasts are lower than A_2B_R, explaining perhaps a lower interaction with left atrium fibrosis modulation [170]. Further studies focusing on A_2B_R activation or blockade in AF and on atrial fibrosis are needed.

## 5. Association between Atrial Fibrillation Risk Factors and the Adenosinergic System

Dysregulation of the autonomous nervous system are characterized by an excessive sympathetic activation, and a diminished parasympathetic influence is central to the pathogenesis of cardiovascular diseases, including heart failure, hypertension and AF [141]. Combined sympatho-vagal activation reflects the equilibrium between the release of epinephrine/norepinephrine, which activate the adrenergic receptors, and the release of acetylcholine, which induces the activation of the muscarinic receptors.

Atrial sympathetic innervation is controlled by adrenergic receptors, which are divided into three major subfamilies: α1-, α2-, and β-adrenergic receptors. They are all coupled to different classes of heteromeric G proteins. The α1 and α2 receptors are coupled to Gq/11 and Gi/o, respectively. β-Adrenergic receptors are coupled to Gs. The α1 and α2 receptors induce a strong extracellular ATP release, while the β-adrenergic receptors activate cAMP production [9]. The activation of the β-adrenergic receptors by isoproterenol leads to the phosphorylation of nonjunctional RyR2 and L-type Ca^2+^ channels (LTCC) and is responsible for large increases in the Ca^2+^ flux [78]. Interestingly, after the use of an A_2A_R agonist, the increase in the heart rate was attenuated by a β-blocker. Thus, Wragg et al. demonstrated a direct activation of the sympathetic nervous system by A_2A_R stimulation [171].

In parallel, atrial parasympathetic innervation is controlled by muscarinic receptors (M_2_Rs). As with A_1_R stimulation, M_2_R activation leads to the opposite effect on β-adrenergic stimulation. The M_2_R stimulation activates inhibitory G proteins, subsequently reduces the activity of the HCN and LTCC and leads to decreased automaticity and conduction velocity in the nodal cell. M_2_R also activates the GIRK channel responsible for hyperpolarization [172].

Sympathovagal activation is a strong modulator of AF. Interestingly, hypertension, sleep apnea and heart failure are known to induce sympathetic tone activation and specific atrial remodeling [1,173,174,175,176]. Moreover, all three AF risk factors also induce specific adenosinergic system remodeling. Especially in patients suffering from essential hypertension, an A_2A_R overexpression was described in PBMCs [139]. Because of the vasodilator effect of A_2A_R, it was hypothesized that the A_2A_R overexpression was a compensatory mechanism of high blood pressure [176]. However, the chronic release of adenosine in the peripheral cardiovascular system during high blood pressure may also induce atrial remodeling of adenosine receptors. In the same manner, in sleep apnea or heart failure, the associated hypoxia may contribute to the atrial adenosinergic system’s remodeling and induce a pro-arrhythmogenic environment.

To go further, a recent study confirmed that vagus nerve stimulation promotes atrial pro-arrhythmogenicity, mainly through GIRK channel activation [172]. Interestingly, GIRK activation is enhanced in persistent AF [177], which can promote APD shortening even in the absence of vagal stimulation [178].

As the A_1_R stimulation induces GIRK activation, the question remains whether A_1_R remodeling and its activation by adenosine can also contribute to arrhythmogenic effects through a similar pathway.

The stimulation of the adrenergic and parasympathetic tone alone or combined can predispose to the AF onset [179]. Sequential combined stimulations had a synergic effect rather than vagal or sympathetic drive alone [179]. Interestingly, heart rate variability analyses before the occurrence of AF showed that AF patients have specific sympathetic and parasympathetic patterns [180]. Furthermore, prolonged atrial pacing in dogs induced sympathovagal activation and increased the risk of AF. Interestingly, the cryoablation of both autonomic nerves prevents the occurrence of AF [181]. However, the exact interaction between the adenosine receptors and cardiac autonomic innervation, as facilitator or inhibitor, is still unclear.

## 6. Conclusions

The remodeling of the adenosinergic system has been described in many cardiovascular diseases; however, it is of note that the repetition of pathophysiological characteristics by exogenous adenosine injection have been described only in syncope [182] and atrial fibrillation [144], which underlies a direct causal role of adenosine in AF. This review illustrates how the adenosinergic system interferes with all steps of the Coumel triangle to promote the initiation and maintenance of AF. Indeed, adenosine receptors’ activation promotes the occurrence of delayed depolarization, all the more so as A_2A_Rs are overexpressed. Adenosine, through A_1_R activation, shortened the action potential duration and induced resting membrane potential hyperpolarization, which both initiates and maintains AF by promoting pulmonary vein firing, stabilizing the AF rotors and allowing for functional reentry. Moreover, A_2B_R has been associated with atrial fibrosis homeostasis, and the adenosinergic system can modulate the autonomous nervous system and is associated with AF risk factors (heart failure, arterial hypertension and sleep apnea).

The question remains whether adenosine release would be a cause or a consequence of AF. On the one hand, the release of adenosine could be a consequence of increased energy requirements related to AF risk factors, which then secondarily enhance AF. On the other hand, adenosine release could be just a consequence of the cardiac hypoxia associated with AF.

## 7. Future Directions

As illustrated in this review, multiple electrophysiological modulations secondary to adenosine receptor activation and overexpression may coexist as part of the initiation, recurrence or maintenance of AF. To go further, continuous monitoring of adenosine plasma levels could help to understand whether AF or adenosine is the chicken or the egg.

It is also hypothesized that the vicious circle of AF initiation and maintenance may also be driven by the modulation of adenosine receptors’ expression. Until now, the antiarrhythmic action of adenosine is still debated, and the results of studies evaluating the antiarrhythmic actions of adenosine agonists or antagonists are controversial. To overcome this barrier, research efforts in AF could explore other pharmacological approaches, such as biased agonist at adenosine receptors. The specific modulation of adenosine receptor signaling and activity may then play a protective antiarrhythmic role in AF.

## Figures and Tables

**Figure 1 biomedicines-10-02963-f001:**
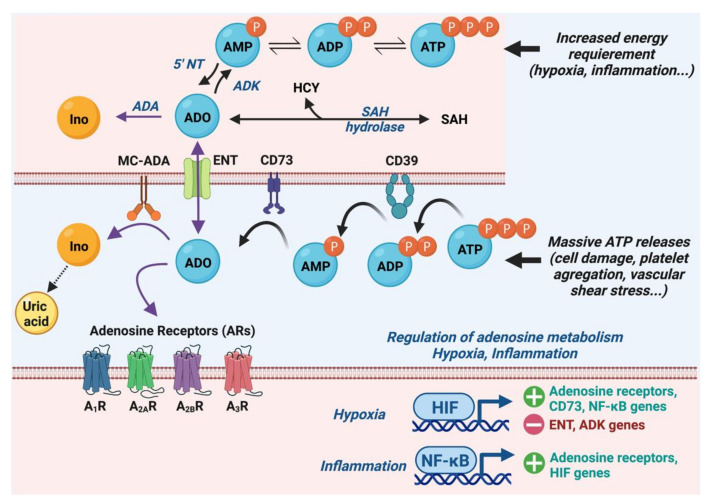
Adenosine metabolism. The sources of intracellular adenosine (ADO) come partly from the methionine cycle by the hydrolysis of S-adenosylhomocysteine (SAH) and mainly from the ATP dephosphorylation cascade when cellular energy requirement increases. Massive extracellular release of ATP produces adenosine by the consecutive actions of ectonucleoside triphosphate diphosphohydrolase-1 (CD39) and ecto-5’-nucleotidase (CD73). Adenosine can bind to adenosine receptors (ARs), but it is also reuptake by equilibrative nucleoside transporters (ENT_1–2_) and rephosphorylated by adenosine kinase (ADK). Adenosine is catabolized by adenosine deaminase (ADA) into inosine (Ino) and, finally, joins uric acid metabolism. Hypoxia and inflammation increase the expression of adenosine receptors and the production of extracellular adenosine: HIF promotes the transcription of adenosine receptors and CD73 and NF-kB genes and represses ENT and ADK genes; NF-κB promotes the transcription of adenosine receptors and HIF genes. ATP: adenosine triphosphate; ADP: adenosine diphosphate; AMP: adenosine monophosphate; HCY: homocysteine; HIFs: hypoxia-inducible transcription factors; MC-ADA: mononuclear cell ADA; NF-kB: nuclear factor-kappa B.

**Figure 2 biomedicines-10-02963-f002:**
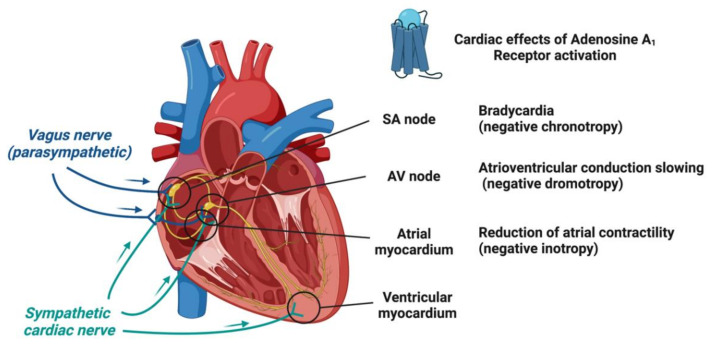
Cardioprotective effects of adenosine A_1_ receptors’ activation. Stimulation of the A_1_ receptors induces bradycardia (negative chronotropy) on a sinoatrial node (SA node), atrioventricular conduction slowing (negative dromotropy) on an atrioventricular node (AV node) and reduced atrial cardiomyocyte contractility (negative inotropy).

**Figure 3 biomedicines-10-02963-f003:**
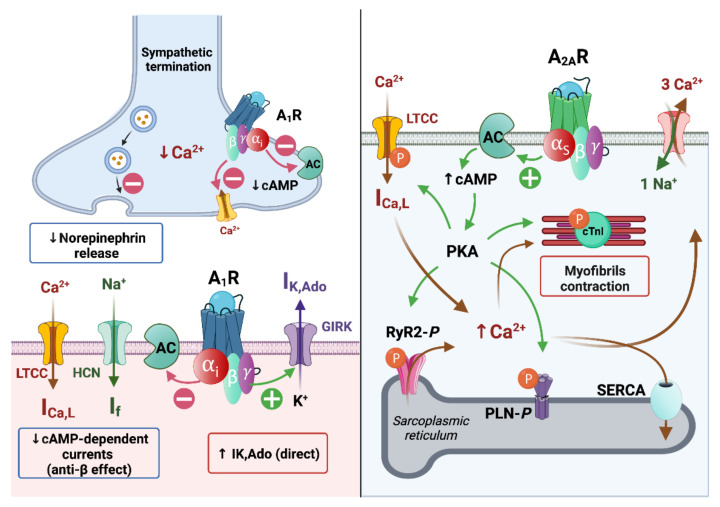
Adenosine receptors signaling (**right**: A_1_R; **left**: A_2A_R). Prejunctional A_1_R activation limits the norepinephrine release. Complementarily, A_1_Rs reduce the increased adrenergic-induced cAMP elevation and the cAMP-dependent currents (I funny, ICa,L). The main effect of A_1_R activation is mediated through GIRK-induced outward K+ current (IK,Ado). A_2A_R activation increases cAMP and, consequently, activates the PKA. PKA phosphorylates L-type Ca^2+^ channels (LTCC) inducing an inward Ca^2+^ current (ICa,L). Ryanodine receptors’ (RyR2) phosphorylation by the PKA allows for the Ca^2+^-induced Ca^2+^ release from the sarcoplasmic reticulum. Cardiac troponin I phosphorylation and cytosolic Ca^2+^ sparks trigger the myofibrils’ contraction during systole. During diastole, the phosphorylation of phospholamban (PLN) permits the SERCA to pump Ca^2+^ into the sarcoplasmic reticulum. The Na^+^: Ca^2+^ exchanger extrudes Ca^2+^ allowing for diastolic relaxation. AC: adenylyl cyclase; cTnI: cardiac troponin I; PKA: protein kinase A.

**Figure 4 biomedicines-10-02963-f004:**
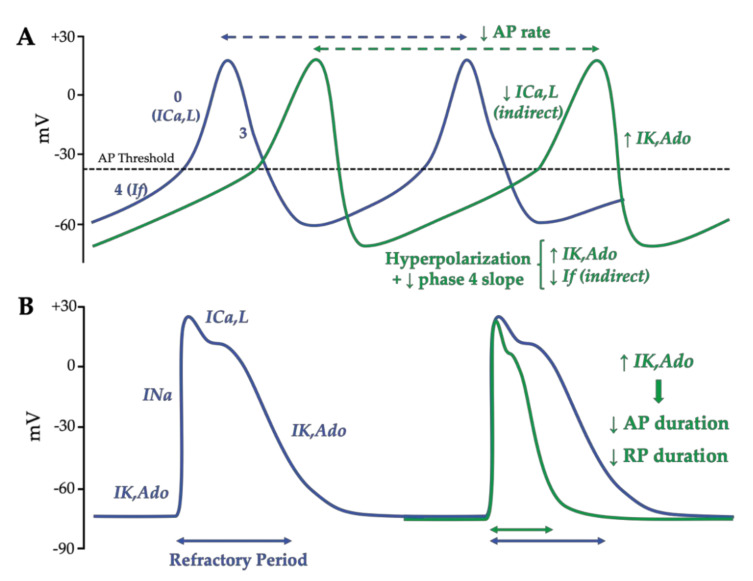
Illustrations of the effects of adenosine on action potential in a sinoatrial cell (**A**) and atrial cardiomyocyte (**B**). In a sinoatrial node, adenosine increases the outward IK,Ado current, which induces hyperpolarization and reduces the slope of the diastolic depolarization. Complementarily, adenosine reduces indirectly the inward Na^+^ pacemaker current (If) and the inward Ca^2+^-dependent depolarization through L-type Ca^2+^ channels (ICa,L). Adenosine incudes negative chronotropy. On an atrial working myocardium, adenosine increases the outward K^+^ current (IK,Ado) and shortens the action potential duration and the refractoriness, thereby adenosine can reduce atrial contractility. Dotted arrows correspond to the corresponding RR intervals. Solid arrows correspond to the corresponding action potential refractory period.

**Figure 5 biomedicines-10-02963-f005:**
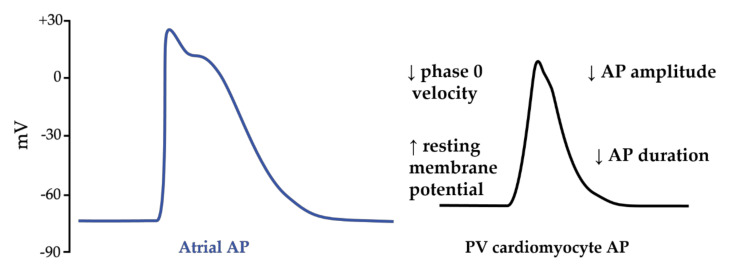
Schematic illustration of differences between action potentials (APs) in atrial cardiomyocytes and in pulmonary vein (PV) cardiomyocytes. Action potential of PV cardiomyocytes have a more depolarized resting membrane potential, a lower AP amplitude, a smaller maximum phase 0 upstroke velocity and a shorter action potential duration compared to atrial AP.

**Figure 6 biomedicines-10-02963-f006:**
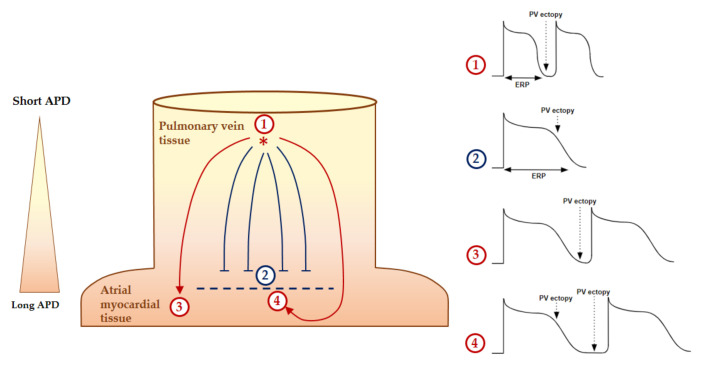
Formation of a functional reentrant circuit due to the heterogeneity of the atrial potential duration at the pulmonary vein (PV) entrance. The delay after depolarization in area no. ① initiates a PV ectopy within the PV sleeve, where the action potential duration (APD) is shorter, with a subsequently shorter effective refractory period (ERP) and propagates toward the PV junction where the APD is longer. The ectopic beat blocks are in area no. ② because of the refractory tissue. However, the impulse propagation continues laterally until the APD tissue is out of the refractory time (area no. ③.). Finally, the PV ectopic beats move around the functional refractory area and then across the previous refractory region of the block in area no. ④, which is now out of the refractory time. The dotted line represents the functional unidirectional refractory area, the solid arrows represent the propagation of the action potential and the star represents the initiation of the ectopic focus.

**Figure 7 biomedicines-10-02963-f007:**
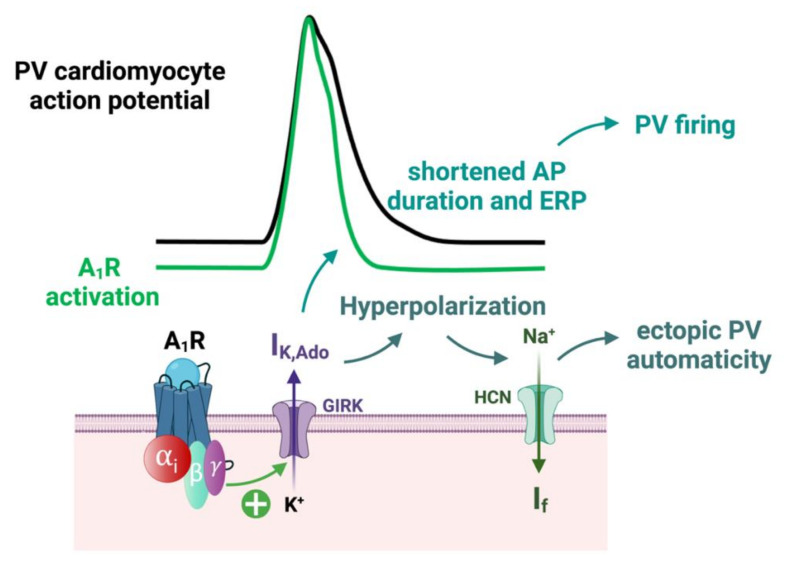
Effects of A_1_R activation by adenosine on pulmonary vein (PV) action potential (AP). AP: action potential; ERP: effective refractory period; PV: pulmonary vein. In PV sleeves of AF patients, A_1_R activation opens GIRK channels, which are responsible for an outward potassium current (IK,ADO). This current shortens the atrial AP duration and effective refractory period (ERP). This enhanced PV firing follows PV ectopy. Furthermore, the outward potassium current is also responsible for a resting membrane potential hyperpolarization which reactivates the If current, increasing PV automaticity.

**Figure 8 biomedicines-10-02963-f008:**
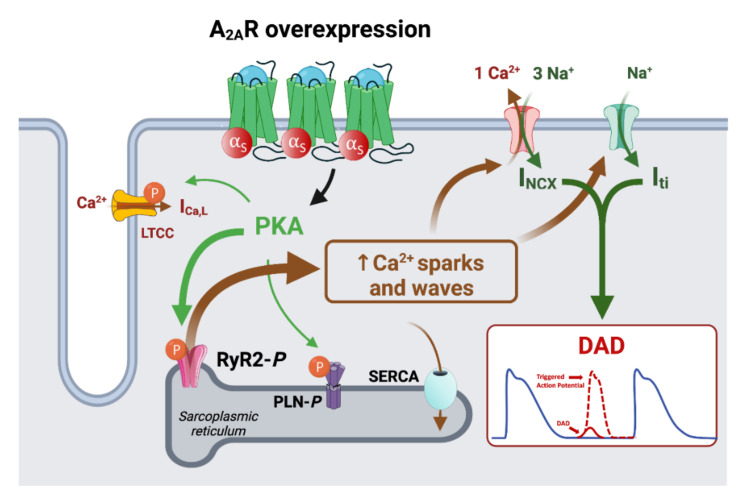
Remodeling of the calcium handling associated with A_2A_R overexpression in AF. Atrial remodeling in atrial fibrillation is associated with the overexpression of A_2A_R. This is responsible for an increased phosphorylation state of ryanodine receptors into dyads, which can induce diastolic Ca^2+^ leaks without modulating the L-type Ca^2+^ current. In parallel, the cytosolic Ca^2+^ overload activates the Na^+^-Ca^2+^ antiporter exchanger to remove Ca^2+^ from cells with an accepted stoichiometry of 3Na^+^:1Ca^2+^. Consequently, the associated cytosolic Ca^2+^ diastolic overload and the increased cation inward currents can induce delayed afterdepolarizations, which can be sufficient to meet the threshold cell membrane voltage and to trigger an action potential.

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
