# Peer review of "Adenosine and Adenosine Receptors: Advances in Atrial Fibrillation"

_biomedicines, 2022, doi:10.3390/biomedicines10112963_

Round 1
Reviewer 1 Report
This review article by Maille et al. focused on the role of adenosine and adenosine receptors on advances in atrial fibrillation (AF). Authors described adenosinergic system signaling and the pathophysiology of AF, arrhythmogenic effects of adenosine, and the association between AF risk factors and the adenosinergic system disturbance. As authors mentioned, adenosine and adenosine receptors have reportedly pivotal role on the development of AF but it remains controversial whether the adenosine release and the adenosine receptors activation are a cause or consequence of AF. Therefore, the concept of this review article to clarify it is valuable and overall contents seem satisfactorily described well and educational. I do not have any concern to be resolved.
Author Response
Dear reviewer,
Thank you for your reviewing.
Please find attached a modified version of the manuscript.
Best regards,
Baptiste MAILLE (corresponding author)
Reviewer 2 Report
I have read the manuscript. It is an interesting work and well written. I have few suggestions:
- the format of the paper should be revised according with the journal Guidelines.
- the abstract is less informative that the entire paper. I suggest revising the abstract.
Author Response
Dear reviewer,
Thank you very much for your reviewing.
According to your comments we modified the initial version of the manuscript.
Please find modification detailed below, in the new submitted version.
"the format of the paper should be revised according with the journal Guidelines"
- We divided the last section (initially 6.conclusion and perspectives) in two different sections 6. Conclusion and 7. Future direction
- We modified the back matter section:
- We started with funding rather than author contributions
- then we added a conflict of interest section
"The abstract is less informative that the entire paper. I suggest revising the abstract."
Thanks to your commentary we improved details of the abstract, according with the graphical abstract:
- Initial abstract"Adenosine receptors activation can lead to AF through several mechanisms including shortening of the action potential duration and effective refractory period, hyperpolarization of the resting membrane potential, enhancement of delayed after depolarization and modulations of the atrial fibrosis and the autonomic nervous system.",
- Has been replaced by " Indeed, activation of overexpressed A2A receptors can promote triggers through an increased occurrence of delayed after depolarization. Activation of A1 receptors by adenosine can shorten action potential duration and induce resting membrane potential hyperpolarization which both initiate and maintain AF by promoting pulmonary vein firing, stabilizing AF rotors and allowing functional re-entry. Moreover, A2B receptors have been associated with atrial fibrosis homeostasis. Finally, the adenosinergic system can modulate the autonomous nervous system and is associated with AF risk factors.
- Rest of the abstract have has not been modified.
Best regards,
Dr Baptiste MAILLE (Corresponding author)